# Identification of Biomarkers Associated with Liver Metastasis Progression from Colorectal Cancer Using Exosomal RNA Profiling

**DOI:** 10.3390/cancers14194723

**Published:** 2022-09-28

**Authors:** Soohyeon Lee, Young Soo Park, Jwa Hoon Kim, Ah Reum Lim, Myung Han Hyun, Boyeon Kim, Jong Won Lee, Saet Byeol Lee, Yeul Hong Kim

**Affiliations:** 1Division of Medical Oncology and Hematology, Department of Internal Medicine, Korea University College of Medicine, Seoul 02841, Korea; 2Cancer Research Institute, Korea University College of Medicine, Seoul 02841, Korea; 3Brain Korea 21 Plus Project for Biomedical Science, Korea University College of Medicine, Seoul 02841, Korea; 4Department of Biomedical Sciences, Korea University College of Medicine, Seoul 02841, Korea

**Keywords:** liquid biopsy, exosomal RNA, metastatic colorectal cancer, liver metastasis, therapy resistance

## Abstract

**Simple Summary:**

Exosomes are a class of extracellular vesicles released by cancer cells that play important roles in cancer progression and therapy resistance. Using RNA sequencing in plasma exosomes from patients with metastatic colorectal cancer (mCRC), we investigated exosomal RNA expression profiles between initial recurrence and cancer progression after metastasectomy. The differential expression level of the exosome RNA was enriched in samples from patients with metastases to the liver, multiple metastatic sites and large tumor burden. In terms of liver metastasis, the interferon-α response gene set was enriched in a tumor burden of ≥1 cm^3^, and CXCL10, CXCL11 and SAMD 9 were highly expressed in the exosomal RNA which was validated using GSEA. High CXCL10 expression was associated with shorter progression-free survival (PFS), but CXCL11 and SAMD 9 were not correlated with PFS. Exosomal CXCL10 RNA could be a novel biomarker for liver metastasis from mCRC and a potential target for the prevention and treatment of these patients.

**Abstract:**

This study aimed to identify novel biomarkers for metastatic colorectal cancer progression using exosomal RNA expression profiling. The exosomal RNA expression profiles of 54 patients with mCRC were investigated. Exosomal RNA profiling was performed at the time of relapse immediately before metastasectomy and cancer recurrence or progression after metastasectomy. The up- and down-regulated RNA expression profiles were screened and analyzed using H-cluster, principle component analysis and gene ontology. The tissue expression profile of the liver metastases was compared with the GSE 41258 set using GSEA tools. We identified two distinctive biological process gene sets (IFNA and PCDB families) related to metastatic progression. The interferon-α response gene set was enriched, especially when the tumor volume was ≥1 cm^3^. CXCL10, CXCL11 and SAMD 9 mRNA were highly expressed in the plasma exosome samples of patients with mCRC to the liver. Furthermore, high expression of CXCL10 but not CXCL11 or SAMD9 was associated with a poor prognosis and shorter progression-free survival. Conclusions: Cancer-derived exosomal CXCL10 may be a novel biomarker for liver metastasis of mCRC and a potential target for the prevention and treatment of mCRC with liver metastasis.

## 1. Introduction

Colorectal cancer (CRC) is the third most common cancer in Korea and worldwide [1]. Among patients diagnosed with CRC, 20–25% have metastatic CRC (mCRC), and 40% present with recurrence after previously treated localized disease [2]. mCRC remains incurable in most patients, with a 5-year survival rate of less than 20%. The major reason for poor outcomes is therapy resistance. Despite the development of chemotherapy and targeted agents, nearly all patients treated with systemic chemotherapy eventually develop resistance and ultimately succumb to the progression of metastatic disease. Understanding the mechanisms by which cancer cells evade treatment could unlock novel therapeutic strategies to overcome drug resistance and improve survival.

Liquid biopsy in cancer is an attractive option to improve early detection and treatment stratification as well as to monitor recurrence and detect residual disease [3]. In the context of liquid biopsies, the advent of exosome research shows promise in the determination of the complexity of cancer [4]. Exosomes are 30–150 nm phospholipid bilayer-enclosed vesicles surrounding a small amount of cytosol and are recognized as key autocrine and paracrine mediators of cell-to-cell communication through the transportation of bioactive molecules such as mRNA, proteins and microRNAs (miRNAs) [5]. Functionally, exosomes have been shown to influence the tumor microenvironment as vehicles, harboring a diverse repertoire of molecular cargo that is shielded from the degradation in circulation and that is representative of their originating cells. Therefore, the quality, diversity and tumor-specific nature of exosomal DNA and exosomal RNA provides a potentially favorable alternative to cell-free nucleic acids for comprehensive high-resolution tumor profiling. The exosomal proteome may reflect not only aberrant cancer signaling but also the potential stromal response of tumors. A key advantage of exosomal RNA is the preserved quality of transcripts, which alters the transcriptome of the recipient cell, thereby accelerating the progression of cancer. Exosomal RNAs facilitate drug resistance through the efflux of intracellular drugs, the inhibition of apoptosis, DNA-repair enhancement and the transport of drug efflux pumps to drug-sensitive cells. Using tumor tissues and serum exosomes collected before and after chemotherapy, specific gene expression can be altered in the tissues and exosomes of progressive/stable disease groups rather than in the partial/complete response group. Next generation RNA sequencing (RNA-seq) has been employed to explore the expressions of genes that may regulate the drug sensitivity of cancer cells from formalin-fixed paraffin-embedded [6] exosomes in various types of cancers [7]. However, the mechanism underlying the association between RNA profiling dynamics and mCRC progression remains to be fully elucidated.

To gain better insight into the key genes and pathways involved in therapy resistance and progression in mCRC, a bioinformatic analysis of the exosomal RNA in mCRC with oligometastasis and GSE 41258 gene expression profiles was conducted to identify potential key genes and pathways associated with metastatic progression after metastasectomy.

## 2. Materials and Methods

### 2.1. Study Design and Participants

We prospectively enrolled patients with mCRC who underwent surgical resection of the primary tumor and distant metastasis with curative intent at the Korea University Anam Hospital between March 2017 and February 2020. Patients aged >19 years with stage IV CRC and oligometastasis who were planning metastasectomy with curative intent were included. The decision regarding the chemotherapy timing and regimen was made by the treating physicians based on the current mCRC treatment guidelines. In this cohort, paired blood samples were obtained from 54 patients at the time of premetastasectomy and cancer progression after metastasectomy. The samples of all participants were collected with informed consent. This study was approved by the Institutional Review Board of the Korea University Anam Hospital (2017AN0070).

### 2.2. Sample Preparation and the Isolation and Identification of Exosomes

Plasma was isolated within 1 h of blood collection by centrifugation of whole blood at 2500 rpm for 7 min at room temperature. The supernatant was transferred to a fresh tube and centrifuged at 13,000 rpm for 10 min at 4 °C. The supernatant was taken as plasma and aliquoted into a cryo-tube for storage at −80 °C until use.

### 2.3. Exosomal RNA Extraction and Sequencing

Exosomal RNA was extracted using Qiagen exoRNeasy (Qiagen, Germantown, MD, USA) according to the manufacturer instructions. The concentration and quality were measured using a tape station (Agilent, Santa Clara, CA, USA) and a high sensitivity RNA screen tape. RNA samples were stored at −80 °C until sequencing. A low input of 100 pg of RNA samples was used for amplification using the Ion AmpliSeq Transcriptome Human Gene Expression Kit (Thermo Fisher Scientific, Van Allen Way, Carlsbad, CA, USA). The RNA libraries were prepared following manufacturer instructions. Briefly, for first strand cDNA synthesis, 100 pg of low input, high-quality RNA was used to reverse the transcribe and amplify the targets with maximum PCR cycles. The purified amplicons were partially digested to remove primer sequences, ligated to adapters with barcode sequences and purified. The final amplicons were quantified by qPCR for the undiluted Ampli-Seq Transcriptome library to normalize each library to 100 pM. RNA sequencing was performed using an Ion Torrent PGM system with a target probe of 25,000 primers, following the manufacturer recommendations. After sequencing, the average read of the samples was 7,195,306 and of on-target reads was 73.76%.

### 2.4. Data Processing and Analysis

The sequencing results were normalized as fragments per kilobase of exon per million (FPKM), and each gene transcript was normalized to beta-actin level. Expression profiles were compared between baseline and cancer progression. Five-fold increases and decreases in differential expression genes (DEGs) for each individual were listed as log ratios. A total of 7787 genes were assigned to hierarchical clustering for screening highly contributing genes using R Bioconductor, and principal component analysis (PCA) was conducted using SPSS software. The factor scores for grouping and significance gene lists were used as the significance gene lists. For further functional enrichment analysis of DEGs in mCRC with oligometastasis, gene ontology (GO) enrichment analysis, including the biological process (BP) and the Kyoto Encyclopedia of Genes and Genomes (KEGG) pathway were performed using the Database for Annotation, Visualization and Integrated Discovery (DAVID, http://david.abcc.ncifcrf.gob/ (accessed on 10 November 2021)) web-based platform. Gene set enrichment analysis (GSEA) from the Broad Institute (software.broadinstitute.org/gsea/index.jsp (accessed on 6 December 2021)) was used to cluster significant gene sets associated with given annotated terms. In the present study, mCRC samples with liver metastasis were analyzed using GSEA with the annotation of ‘hallmark gene sets’. A significant cut-off value was defined as *p* < 0.05. Candidate gene families were analyzed using MeV software for H-clustering to visualize clinicopathological information and gene expression. The RNA-seq and validation workflows are presented in Figure 1.

### 2.5. Statistical Analysis

Statistical analysis was performed using the R 4.0.2 software. DAVID gene ontology was considered statistically significant with *p* < 0.05 and FDR < 0.25.

## 3. Results

### 3.1. Patient Characteristics

This study enrolled consecutive patients with mCRC who had relapsed with oligometastasis and then underwent metastasectomy with or without chemotherapy. The median age of the patients was 57 years, and more than half of the patients were male (*n* = 31/54, 57%). Of the 54 patients who underwent metastasectomy with curative intent, 42 received systemic chemotherapy before metastasectomy: cytotoxic chemotherapy only (*n* = 4); bevacizumab-containing chemotherapy (*n* = 24); cetuximab-containing chemotherapy (*n* = 14); and 12 patients underwent metastasectomy immediately without chemotherapy. The median progression-free survival (PFS) after metastasectomy was 374 days. The recurrence sites were the liver (*n* = 16, 30%), lung (*n* = 15, 28%), liver and lung (*n* = 20, 37%) and others (lymph node and peritoneum, *n* = 3, 6%). The sum of the metastatic tumor burden was smaller <1 cm^3^ in 24 patients (46%) and larger than 1 cm^3^ in 30 patients (54%). The clinical data of patients are summarized in Table 1.

### 3.2. Exosomal RNA Expression Patterns Analysis Using H-Clustering and PCA

We obtained the gene expression profiles of serum exosomes by mapping RNA sequencing reads and identified 7877 expressed genes. To obtain the differential RNA expression profiles in serum exosomes from patients with mCRC whose cancer had progressed, we selected the DEGs by pairwise comparison (fold change, 5; *p* < 0.05) and plotted the data on heatmaps (Figure 2).

H-clustering analysis revealed a difference in the expression profiles according to factors such as metastasis location and tumor volume. There was no relationship between the treatment groups and expression profiles.

We further analyzed the expression of these genes using PCA so that the top three principal components and the resulting discriminating exosomal RNA variables displayed on the hierarchical clustering heatmaps represented 80–100% of the variance in each dataset. The main contributors to differences between patients with liver, lung and multiple metastasis were demonstrated by the PCA (Figure 3), in which the first two components together accounted for ~75% of the data variation. The metastatic sites were clearly categorized, although PC1 and PC2 explained 17% and 22%, respectively.

### 3.3. Gene Oncology (GO) Analysis of Exosomal RNA Expression

As PC1 was a major component in dividing patients according to second metastases sites, we sub-listed the top and bottom 1000 genes by the component factor scores and applied these genes to an ontology tool using DAVID. The interferon alpha (INFA) family is involved in the positive regulation of the serine phosphorylation of the STAT protein, natural killer cell activation involved in immune response, T cell activation involved in immune response, B cell proliferation, response to exogenous dsRNA, regulation of the type I interferon-mediated signaling pathway, B cell differentiation and humoral immune response (*p*-value < 0.01, Benjamini < 0.05). In addition, the protocadherin beta (PCDB) family is commonly involved in chemical synaptic transmission, calcium-dependent cell–cell adhesion via plasma membrane cell adhesion molecules and synaptic assembly (Figure 4a). The RIG-I-like receptor pathway is closely related to type I and II interferons and induces the expression of interferon alpha and beta genes (Figure 4b). The expression patterns and patient clusters showed that not only the liver and multiple metastases but also the high tumor volume was dependent on each gene level (Figure 4c).

### 3.4. Validation of Tumor RNA Expression Using GSE 41258

To investigate the exosomal expression of RNA related to drug resistance genes, we performed GSEA. We processed the GSE 41258 data set to compare exosomal RNA expression with tumor tissue RNA expression profiles. Genes up-regulated from the patients with liver metastasis were enriched terms such as interferon-α response, apical junction, angiogenesis, myogenesis, TNF-α signaling via NFκB and inflammatory response. Interferon-α response was a common enrichment hallmark term (Figure 5a,b). Among the INFA response-enriched genes, SAMD9, CXCL10 and CXCL11 genes overlapped in the exosomes; tumor tissue overlapped from the core enrichment genes (Figure 5c). A high expression of CXCL10 was related to a poor prognosis with short PFS, but CXCL11 and SAMD 9 were not correlated with PFS (Appendix A).

## 4. Discussion

Although the survival of patients with mCRC with oligometastasis has significantly improved with advances in surgical techniques and effective chemotherapy regimens, 75% of patients who undergo metastasectomy develop recurrence within 18 months of surgery [8,9,10]. Recurrence occurs even after metastasectomy with or without doublet or triplet chemotherapy because drug-resistant clones can be seeded in other organs through exosomes. Cancer cells first shed exosomes into the blood stream at the initiation step of metastasis; the exosomes travel to distant organs where they settle [11,12]. Exosomes are also potential candidates for distinguishing CRC patients with and without chemoresistance.

In this study, we revealed differential expression profiles of RNAs in plasma exosomes from patients with liver metastasis or a high tumor volume. Our data demonstrate the complexity of gene expression profiles in terms of mRNA and differences. To examine the direct role of exosomes in mCRC with liver metastasis, this study aimed to identify a novel biomarker that could indicate tumor progression using exosomal RNA-seq data. We identified two distinct biological process gene sets (IFNA and PCDB) related to liver metastasis in patients with mCRC. INFA up-regulation resulted in the enrichment of immune-response pathways which could be related to liver metastasis and acquired chemo-resistance. The cluster of the PCDB family that is related to cancer cell progression and distant metastases is also required for therapy resistance [13,14,15]. However, the elucidation of the role of the individual PCDB family members needs further mechanistic studies. Exosomal mRNA analysis showed that CXCL10, CXCL11 and SAMD9 were highly expressed in CRC patients with liver metastasis, which was validated by tumor RNA sequencing using GSEA. A high expression of CXCL10 was related to shorter PFS, but the expression of CXCL11 and SAMD9 was not related to PFS based on the exosomal RNA profiling.

CXCL10 is a member of the CXC chemokine family which binds to the CXCR3 receptor to exert its biological effects. CXCL10, along with other CXC chemokines, binds to G-protein coupled receptors and induces a wide spectrum of biological and physiological activities [16]. CXCL10 is transcriptionally regulated in response to external stimuli such as IFN-γ and lipopolysaccharides (LPS) by a region of 230 nucleotides upstream from the transcriptional start site [17].

CXCL10 is involved in chemotaxis, the induction of apoptosis, the regulation of cell growth, the mediation of angiostatic effects and metastasis [18,19] and has been associated with drug resistance in several studies [20,21,22]. CXCL10 signaling may induce an immunosuppressive niche allowing cancer cells to be accepted in various metastatic organs as well as in the immunosuppressive tumor microenvironment [23,24]. We suggest that CXCL10 differs from other chemokines by its ability to restrain tumor growth and enhance antitumor immunity.

There are several limitations to the use of exosomal RNAs as predictive biomarkers. First, the purity of exosomes is a major concern, and the presence of contaminating proteins and RNAs in exosomal preparations may compromise the accuracy of exosome-based diagnosis. Second, there is high heterogeneity in the vesicles depending on how they are purified. Third, GSEA data were used for indirect validation using exosomal RNA analysis as tumor RNA sequencing was not performed. Considering the dual effect of CXCL10 on cell growth, additional validation studies are needed for the relevance of CXCL10 exosomal RNA to treatment resistance.

Several studies have investigated the relationship between various types of exosome and therapy resistance. A previous study confirmed the enrichment of oncogene miR-21 in the exosomes from CRC-associated fibroblasts (CAFs) in chemoresistance to oxaliplatin, and long noncoding RNA (lncRNA) H19 is highly expressed in the CAFs of CRC patients and with a marked increase with cancer progression. This lncRNA is transferred to CRC cells through exosomes and thus contributes to the chemoresistance to oxaliplatin by inhibiting miR-141 [25]. The studies focusing on the role of macrophages have shown that tumor-associated macrophages (TAMs) play a dominant role in mediating CXCL12/CXCR4-induced liver metastasis of CRC. Several miRNAs (miR-25-3p, miR-130b-3p, miR425-5p), upregulated by the activation of the CLCL12/CXCR4 axis, can be transferred to macrophages via exosomes [26]. Tumor-derived exosomal miR-934 cancer promotes CRC with liver metastasis by regulating the crosstalk between CRC cells and TMAs which reveals a tumor and TAM interaction in the metastatic microenvironment, mediated by tumor-derived exosomes that affect the liver metastasis of CRC [27].

Overall, analysis of exosomal RNAs unveiled some potential biomarkers for the early detection of liver metastasis of mCRC after metastasectomy and provided clues about the possible mechanisms of chemoresistance in the context of an immune-suppressive microenvironment.

## 5. Conclusions

Using the exosomal RNA sequencing method, we analyzed gene expression profiling between two time points, immediately after curative metastasectomy and metastasis progression. Even though patients with liver metastases progression from CRC are extremely heterogeneous, immune evasion with high IFNA response genes can reveal a pivotal role for immune editing in metastatic progression. The high expression of CXCL10 exosomal RNA could be a biomarker to suspect disease progression or therapy resistance.

## Figures and Tables

**Figure 1 cancers-14-04723-f001:**
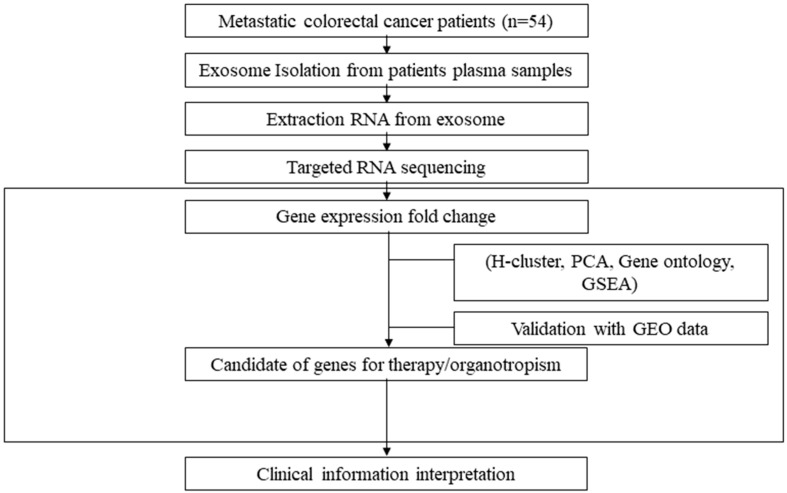
Overview of the plasma exosomal RNA analysis workflow. Abbreviations: PCA, principal component analysis; GSEA, gene set enrichment analysis; and GEO, gene expression omnibus.

**Figure 2 cancers-14-04723-f002:**
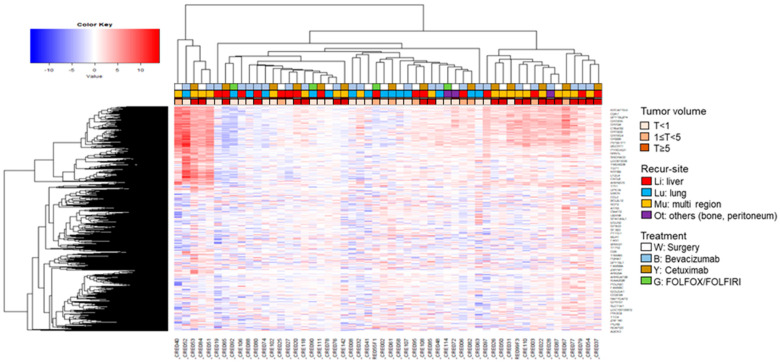
Hierarchical clustering of exosomal RNA expression profiles.

**Figure 3 cancers-14-04723-f003:**
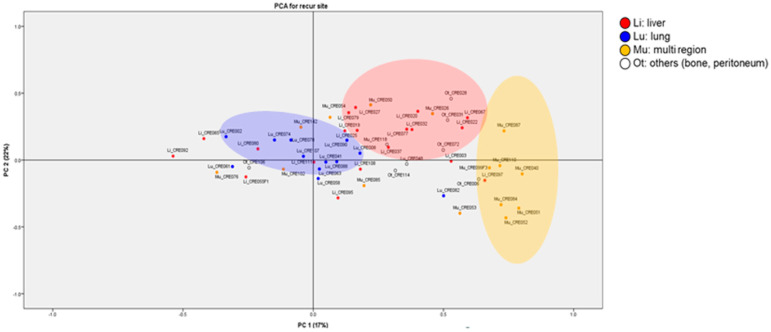
The principal component analysis (PCA) based on detected exosomal RNA levels.

**Figure 4 cancers-14-04723-f004:**
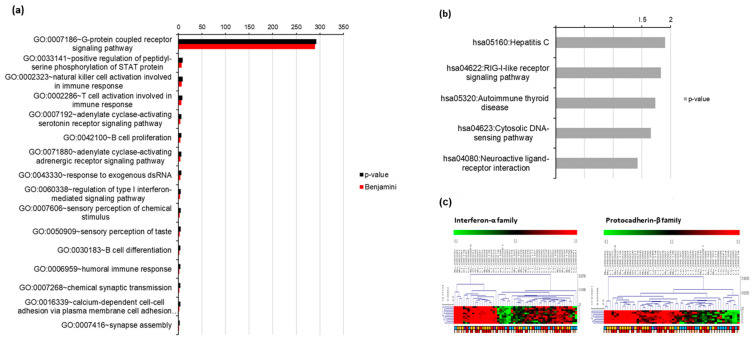
Gene ontology (GO) of exosomal RNA in mCRC patients: (**a**) biologic process, (**b**) KEGG pathway and (**c**) common enriched gene sets. Abbreviations: mCRC, metastatic colorectal cancer; and KEGG, Kyoto Encyclopedia of Genes and Genomes.

**Figure 5 cancers-14-04723-f005:**
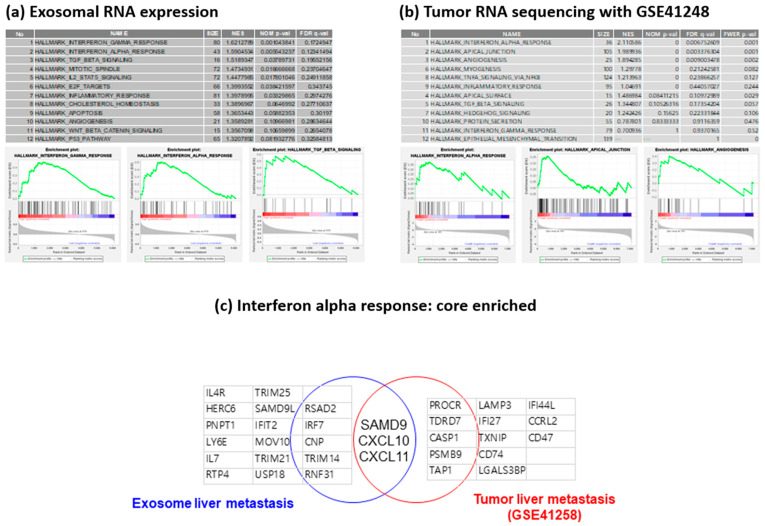
Comparison of RNA expression profiles of mCRC with liver metastasis in (**a**) exosomes, (**b**) GSE41258 and (**c**) common enrichment gene sets in exosome and tumor RNA sequencing.

**Table 1 cancers-14-04723-t001:** Baseline characteristics.

Clinical Variables			Number of Patients (N, %)
Age	<65 years		39 (72.2)
	≥65 years		15 (27.8)
Gender	Male		31 (57.4)
	Female		23 (42.6)
Perioperative therapy	Surgery only		12 (22.2)
	Cytotoxic chemotherapy		4 (7.4)
	Cetuximab-based therapy		14 (25.9)
	Bevacizumab-based therapy		24 (44.4)
Metastatic sites	Liver		16 (29.6)
	Lung		15 (27.9)
	Multiple (only liver and lung)		20 (37.0)
	Others (bone, peritoneum)		3 (5.5)
Metastasis tumor size	<1 cm^3^		25 (46.3)
		Liver	5 (20.0)
		Lung	13 (52.0)
		Multiple	5 (20.0)
		Others	2 (8.0)
	≥1 cm^3^		29 (53.7)
		Liver	11 (38.0)
		Lung	2 (2.9)
		Multiple	15 (51.7)
		Others	1 (3.5)

## Data Availability

The data presented in this study are available on request from the corresponding author. The data are not publicly available due to privacy issues related to the patients.

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
