# Peer review of "Identification of Biomarkers Associated with Liver Metastasis Progression from Colorectal Cancer Using Exosomal RNA Profiling"

_cancers, 2022, doi:10.3390/cancers14194723_

Round 1

Reviewer 1 Report

This study attempts to identify novel biomarkers for metastatic colorectal cancer (mCRC) progression using exosome RNA profiling. Using targeted RNA sequencing analysis from patient samples, the authors identified Cancer-derived exosomal CXCL10 as a novel biomarker for liver metastasis of mCRC. The manuscript is well written and successfully demonstrated the importance of CXCL10 as a biomarker for mCRC.

However, I have a few criticisms regarding the manuscript, which is described below.

1.    The authors explained the importance of CXCL10 exosomal RNA that encodes chemokines but did not describe in detail about its mechanism of secretion and the site of translation. Please include such details in the discussion part.

2.    The author pointed out that the limitations involved in the use of exosomal RNAs as predictive biomarkers. Such limitations can be overcoming by studying the level of Interferon gamma-induced protein 10 (the protein encoded by CXCL10) in the serum. The authors should use a few tumor samples and demonstrate that the protein protein product of CXCL10 exosomal RNA is also present in the serum. Elisa can be used for that purpose. By using such a strategy, the author can provide a perfect validation for their results.

Author Response

  1. The authors explained the importance of CXCL10 exosomal RNA that encodes chemokines but did not describe in detail about its mechanism of secretion and the site of translation. Please include such details in the discussion part.

I re-organized and added some sentences about the meaning of CXCL10 exosomal RNA in the discussion part as below;

CXCL10 is a member of the CXC chemokine family which binds to the CXCR3 receptor to exert its biological effects. CXCL10, along with other CXC chemokines, binds to G-protein coupled receptors and induces a wide spectrum of biological and physiological activities[16]. One of these activities involves the increase of cell growth and proliferation. CXCL10 is involved in chemotaxis, induction of apoptosis, regulation of cell growth, me-diation of angiostatic effects and metastasis[17,18] and have been associated with drug resistance in several studies[19-21]. CXCL10 signaling may induce an immunosuppressive niche allowing cancer cells to be accepted in various metastatic organs as well as in the immunosuppressive tumor microenvironment[22,23]. We suggest that CXCL10 differ from other chemokines by their ability to restrain tumor growth and enhance anti-tumor immunity.

  1. The author pointed out that the limitations involved in the use of exosomal RNAs as predictive biomarkers. Such limitations can be overcoming by studying the level of Interferon gamma-induced protein 10 (the protein encoded by CXCL10) in the serum. The authors should use a few tumor samples and demonstrate that the protein protein product of CXCL10 exosomal RNA is also present in the serum. Elisa can be used for that purpose. By using such a strategy, the author can provide a perfect validation for their results.

Based on your valuable comments, I tried to validate of the meaning of the CXCL10 using ELISA. Unfortunately, the difference between the two time points was not significant, and the CXCL10 concentration showed a tendency to slightly increase when relapsed, but it was not statistically significant. (see the below graph). I thought that the paired plasma samples were not enough to test again with ELISA. Considering the dual properties of CXCL10, it seems necessary to analyze the meaning of CXCL10 in detail in the context of various situations involved in treatment resistance.

Graph image is uploaded as a file. 

Reviewer 2 Report

Lee et al. studied "Identification of biomarkers associated with metastatic colorectal cancer progression using exosomal RNA profiling". The manuscript is interesting and has well written. But this study didn't explain the exosomal RNA profile of normal blood cells for the comparison. This is crucial for explaining the comparative mechanisms of metastatic CRC progression. Do you have a valid reason why you did not do this? Additionally, there are a few clarifications required before accepting for publication. 

The title needs to be revised as this does not tell anything about the source of the blood sample for exosome isolation. 

Delete the subheading of the abstract. This journal follows a non-structured abstract. 

The conclusion does not support your objective. 

Line 64-77 needs appropriate citation for every piece of information. 

Figure 1 caption is wrong. Additionally, use the figure caption below the figure. This is applicable to all figures. 

Doble checks the abbreviation and uses the full form of each abbreviation at the first appearance in the manuscript and does not repeat the full form in other places of the manuscript. 

Results are well written, but PCA didn't write well. Extend the results of PCA from figure 3. 

How do your findings defend the therapy resistance? Need more explanation and can propose a schematic molecular mechanism.

What was the control for comparing the gene expression? 

Author Response

Lee et al. studied "Identification of biomarkers associated with metastatic colorectal cancer progression using exosomal RNA profiling". The manuscript is interesting and has well written. But this study didn't explain the exosomal RNA profile of normal blood cells for the comparison. This is crucial for explaining the comparative mechanisms of metastatic CRC progression. Do you have a valid reason why you did not do this?

Thank you for your comments.

Exosomes represented a novel mode of intercellular communication including normal and cancer cells. During chemotherapy, not only cancer cells but also normal cells are destroyed, so it was thought that exosomes which was found during chemotherapy would have many components of normal cells.

Gene expression is the process by which information from a gene is used in the synthesis of a functional gene product. When the first study was planned, I thought that it was better to compare before and after treatment in the same patient than to compare the normal exosomal RNA expression values. That’s the reason why the DEGs (Differentially Expressed Genes) method was adopted, which is important to understand the biological differences as a whole between the timing of metastasectomy with curative intent and therapy resistant situation.

Additionally, there are a few clarifications required before accepting for publication.

The title needs to be revised as this does not tell anything about the source of the blood sample for exosome isolation.

To clarify the context of this study, I modified the title as below;

Identification of Biomarkers Associated with Liver Metastasis Progression from Colorectal Cancer Using Exosomal RNA Profiling

Delete the subheading of the abstract. This journal follows a non-structured abstract.

I corrected the abstract form with deleting the subheading

The conclusion does not support your objective.

I corrected the conclusion part to support my study objective as below;

Using the exosomal RNA sequencing method, we analyzed gene expression profiling  between two time points such as right after curative metastasectomy and the metastasis 

progression. Even though patients with liver metastases from CRC are extremely hetero-geneous, immune evasion with high IFNA response genes can reveal a pivotal role of immune editing for metastatic progression. The high expression of CXCL10 exosomal RNA could be a biomarker to suspect disease progression or therapy resistance.

Line 64-77 needs appropriate citation for every piece of information.

I cited a Kalluri’s work (reference number 5).

Figure 1 caption is wrong. Additionally, use the figure caption below the figure. This is applicable to all figures.

I corrected Figure 1 caption as below;

Figure 1. Overview of the plasma exosomal RNA analysis workflow. Abbreviations: PCA, principal component analysis; GSEA, gene set enrichment analysis; GEO, gene expression omnibus.

I also repositioned the figure caption below the figure and applied to all figures.

Doble checks the abbreviation and uses the full form of each abbreviation at the first appearance in the manuscript and does not repeat the full form in other places of the manuscript.

I re-checked and corrected the abbreviation and uses the full form or abbreviation in the appropriated way.

Results are well written, but PCA didn't write well. Extend the results of PCA from figure 3.

I described the results of PCA as below;

We further analyzed the expression of these genes using PCA so that the top three principal components and the resulting discriminating exosomal RNA variables dis-played on the hierarchical clustering heatmaps represented 80–100% of the variance in each dataset. The main contributors to differences between patients with liver, lung and multiple metastasis were demonstrated by the PCA (Figure 3), in which the first 2 compo-nents together accounted for ~75% of the data variation. Metastatic sites were clearly cate-gorized although PC1 and PC2 explained 17% and 22%, respectively.

How do your findings defend the therapy resistance? Need more explanation and can propose a schematic molecular mechanism.

Honestly, it is difficult to say that CXCK10 is the one of strong biomarkers that explains treatment resistance.

I tried to validate of the meaning of the CXCL10 using ELISA. Unfortunately, the difference between the two time points was not significant, and the CXCL10 concentration showed a tendency to slightly increase when relapsed, but it was not statistically significant. (see the below graph). I thought that the paired plasma samples were not enough to test again with ELISA. Considering the dual properties of CXCL10, I think that additional studies considering the complex interactions and contexts of multiple factors rather than single factors are needed. (graph was included in the uploaded file) 

What was the control for comparing the gene expression?

As previously answered, I thought that it was better to compare before and after treatment in the same patient than to compare the normal exosomal RNA expression values. That’s the reason why the DEGs (Differentially Expressed Genes) method was adopted.

Round 2

Reviewer 1 Report

This manuscript can be published in the current form.

Reviewer 2 Report

It seems the authors made substantial corrections.